# Outcome of Irrigation and Debridement with Topical Antibiotic Delivery Using Antibiotic-Impregnated Calcium Hydroxyapatite for the Management of Periprosthetic Hip Joint Infection

**DOI:** 10.3390/antibiotics12050938

**Published:** 2023-05-21

**Authors:** Hiroki Wakabayashi, Masahiro Hasegawa, Yohei Naito, Shine Tone, Akihiro Sudo

**Affiliations:** Department of Orthopaedic Surgery, Mie University Graduate School of Medicine, 2-174 Edobashi, Tsu 514-8507, Japan; masahase@clin.medic.mie-u.ac.jp (M.H.); yo-yo@clin.medic.mie-u.ac.jp (Y.N.); s-tone@clin.medic.mie-u.ac.jp (S.T.); a-sudou@clin.medic.mie-u.ac.jp (A.S.)

**Keywords:** periprosthetic hip joint infection, irrigation and debridement, antibiotic-impregnated calcium hydroxyapatite

## Abstract

We assessed the clinical results of irrigation and debridement (I&D) with antibiotic-impregnated calcium hydroxyapatite (CHA) as a novel antibiotic delivery system for the treatment of prosthetic-joint-associated infection (PJI) after total hip arthroplasty (THA). Thirteen patients (14 hips) treated with I&D for PJI after THA at our institution between 1997 and 2017 were retrospectively evaluated. The study group included four men (five hips) and nine women, with an average age of 66.3 years. Four patients (five hips) had symptoms of infection within less than 3 weeks; however, nine patients had symptoms of infection over 3 weeks. All patients received I&D with antibiotic-impregnated CHA in the surrounding bone. In two hips (two cups and one stem), cup and/or stem revision were performed with re-implantation because of implant loosening. In ten patients (11 hips), vancomycin hydrochloride was impregnated in the CHA. The average duration of follow-up was 8.1 years. Four patients included in this study died of other causes, with an average follow-up of 6.7 years. Eleven of thirteen patients (12 of 14 hips) were successfully treated, and no signs of infection were observed at the latest follow-up. In two patients (two hips) for whom treatment failed, infection was successfully treated with two-stage re-implantation. Both patients had diabetes mellitus and symptoms of infection over 3 weeks. Eighty-six percent of patients were successfully treated. No complications were observed with this antibiotic-impregnated CHA. I&D treatment with antibiotic-impregnated CHA produced a higher rate of success in patients with PJI after THA.

## 1. Introduction

After removing implants in patients with prosthetic-joint-associated infection (PJI), clinicians prefer to insert antibiotic-impregnated bone cement in the form of beads or spacers. However, antibiotic-impregnated bone cement is also associated with certain disadvantages, including a short duration of drug release [1], very low release rates [1,2], thermal damage to some antibiotics, reduced biocompatibility with the bone for which additional surgery to remove the bone cement is necessary, and possible bone loss adjacent to loosened spacers. Moreover, two-stage revision arthroplasty using cementless prostheses cannot be used for antibiotic-impregnated bone cement.

Infection after total hip arthroplasty (THA) remains a severe and costly complication [3,4]. A regimen of irrigation and debridement (I&D) with modular component exchange was recently approved for the treatment of acute PJI. In early PJI, debridement with retention of the implant is another attractive treatment option. This procedure reduces morbidity, length of hospital stay, and medical costs compared with one-stage or two-stage revision arthroplasty [5]. Various risk factors associated with treatment success have been described.

Chronic infections follow a persistent and undulating course with frequent exacerbations, are generally not fully responsive to systemic antibiotics, and often recur when treatment regimens are discontinued. Therefore, as the only strategy, topical and systemic antibiotics fail to successfully manage bacteria with a biofilm phenotype and must be combined with other approaches [6].

Antibiotic-impregnated calcium hydroxyapatite (CHA) ceramic has been developed as a new drug delivery system (Figure 1A) [7,8] and has been used for the treatment of PJI after THA. Using this system, all implanted antibiotics are released over a long period, with no trapping of the drug in the composite [7], and high release rates are observed [7]. Additionally, revision arthroplasty as I&D and using two-stage cementless prostheses can be used as novel strategies to control PJI. Using this drug delivery system with CHA, we have treated patients with PJI by replacing the modular components (including the femoral head and acetabular insert) and loosening component and retaining the fixed components.

Accordingly, in this study, we aimed to evaluate the clinical results of antibiotic-impregnated CHA used for I&D treatment of infected hips.

## 2. Results

### 2.1. Pre-Operative Patient Information

Soft tissue was normal in eight patients (nine hips) and abscessed with fistula in four hips (Table 1). One of the patients with fistula developed a productive fistula 4 days before surgery.

Pre-operative hip aspiration and standard microbiologic (aerobic and anaerobic) culture were performed in all patients. In 13 aspirates, the culprit bacterium was identified in all but one sample. In nine patients (10 hips), the microorganisms were Staphylococcus aureus, including methicillin-resistant *S. aureus* (one hip). In two patients, the causative microorganism was Escherichia coli, and Enterococcus faecalis and an unknown microorganism were the causative agents in one patient each. One patient in whom no bacteria were detected showed the presence of purulent fluid and pus during pre-operative hip aspiration and intra-operative findings. We diagnosed PJI based on clinical and intra-operative macroscopic and histological findings. We assumed that the reason for the negative culture was that the patient was taking sulfamethoxazole trimethoprim regularly after kidney transplant (Case11, Table 1).

Four patients (five hips) had symptoms of infection for less than 3 weeks, whereas nine patients had symptoms of infection for more than 3 weeks (Table 1).

### 2.2. Surgery

Open debridement was performed using the previous incision and approach. Synovectomy with all abscessed and necrotic joints and periprosthetic regions was performed via large regions (i.e., the anterior, posterior, superior, and inferior regions), with thorough lavage using antibiotic-laden saline. The implant was dislocated in order to treat all interfaces and was then tested intra-operatively for loosening. The modular components (including the femoral head and acetabular insert) and loosening component were replaced, whereas the fixed components were retained. Ten patients (11 hips) underwent exchange of the liner and head, excluding the first two patients. In two hips (two cups and one stem), cup and/or stem revision were performed with one-stage re-implantation because of implant loosening (Table 2).

All patients underwent I&D with antibiotic-impregnated CHA applied to the surrounding bone (Figure 1A,B). CHA (Bone Ceram P; Olympus Terumo Biomaterials Corp, Tokyo, Japan) in cylindrical shapes was sintered at 1200 °C with a porosity of 30–40%; the diameter of micropores was between 40 and 150 µm. There were two sizes: large and small; the small size (10 mm in diameter and 10 mm in height) was used for I&D. During the operation, the selected antibiotic powder was packed into a central cylindrical cavity (7 mm in diameter and 8 mm-deep) in each porous block, and the cavity was then sealed with a CHA plug (7 mm in diameter and 3 mm in height) (Figure 1A). The typical amount of antibiotics in each ceramic block ranged from 60 to 80 mg. The amount of the antibiotic powder depended on the type of antibiotic used. The types of antibiotics that were used to impregnate the CHA are listed in Table 2. In ten patients (11 hips), vancomycin hydrochloride (VCM) was used as the antibiotic for impregnation in the CHA. We created bone holes on the major trochanter and the acetabulum using an air drill. Antibiotic-impregnated CHAs were implanted as often as possible when there were poor bone stocks because of severe bone loss. We implanted 2–5 antibiotic-impregnated CHAs in bone holes (Figure 1B and Appendix A).

### 2.3. Antibiotherapy

The operative antibiotherapy was systematically initiated following pre-operative hip aspiration. This treatment consisted of intravenous (i.v.) antibiotherapy at the effective dose, which was quickly adapted based on the aspiration findings. CHA was administered as local antibiotherapy, and i.v. antibiotherapy was maintained for 4–9 weeks, followed by oral administration for 6 weeks. Tolerance was satisfactory. 

Treatment outcomes

The average duration of follow-up was 8.1 years (range, 2.9–18.6 years). No patients were lost to follow-up. Four patients included in this study died of other causes, with an average follow-up of 6.7 (4.0–12.5) years before death. There was no evidence of recurrent infections in these four patients.

Eleven of thirteen patients (12 of 14 hips) were successfully treated with no signs of infection at most recent follow-up. In two of thirteen patients (two of 14 hips) for whom treatment failed, infection was successfully treated with two-stage re-implantation with antibiotic-impregnated CHA. Both patients had diabetes mellitus and had symptoms of infection for more than 3 weeks. However, seven of nine patients who showed symptoms of infection for more than 3 weeks were successfully treated with I&D with antibiotic-impregnated CHA. Eighty-six percent of patients were successfully treated by I&D with antibiotic-impregnated CHA. No complications, such as excessive postoperative drainage, erythema, bone damage from friction, or any particle disease, were observed following treatment with this antibiotic-impregnated CHA. However, one patient had a greater trochanteric fracture after PJI surgery. There was no pain or dysfunction in this patient, and he followed up normally. These patients were free of infection at the time of the most recent follow-up. Radiographically, there was no loosening or migration of the components in any of the 13 patients.

### 2.4. Functional Outcomes

Eleven of thirteen patients (12 of 14 hips) were comprehensively rated using JOA hip scores at the most recent follow-up. Overall, the median and mean JOA hip scores were 70 and 69.3 (42–97), respectively. For two hips (two patients) that underwent two-stage revision, the JOA hip scores were 59 and 46.

## 3. Discussion

Total joint arthroplasty is a widely used treatment modality for advanced osteoarthritis of the hip and knee. Although this procedure is highly successful, PJI is a relatively uncommon but devastating complication following total joint arthroplasty. The average reported incidence of PJI is 0.5–2% [9,10,11]. PJI has a negative impact on the patient and can cause significant morbidity and mortality, leading to massively increased healthcare costs [9,12,13].

Primary surgical management strategies for PJI differ among institutions but include I&D with modular component exchange, one-stage revision arthroplasty, and two-stage revision arthroplasty. However, two-stage revision is associated with significant morbidity and mortality and is poorly tolerated by patients. Additionally, when associated with a period without a hip implant, the tissue changes can lead to important functional deficits after re-implantation [14].

In appropriate patients, single-stage revision appears to be associated with similar reinfection rates when compared with two-stage revision with superior functional outcomes [15]. One option for the surgical treatment of early PJI is I&D with modular component exchange, which is considered a less invasive surgical treatment option than revision surgery owing to the preservation of bone stock, shorter duration of the surgical procedure, decreased risk of intra-operative fractures, and faster postoperative rehabilitation [5]. Moreover, I&D offers the benefits of decreased patient morbidity and may decrease healthcare costs compared with two-stage revision arthroplasty; however, the reported outcomes are heterogeneous, concerning, and suggest more detailed considerations. The rates of clinical cure after DAIR are highly variable, ranging from 37% to 88%, with the average success rate being reported at around 50% [16,17,18,19,20,21].

Multiple factors influence the outcomes of I&D. For example, the causative microorganisms, applied antimicrobial regimen, soft tissue envelope, timing of intervention, and duration of symptoms have all been shown to influence the outcome of I&D [17,18,19,20]. Other considerations include host-related factors (the patient’s overall health status, medical comorbidities, and immune status), and implant-related factors (implant stability and fixation) [19].

Moreover, several studies have demonstrated that the I&D failure rate increases when the infection persists for a longer period of time [17,20]. In chronic infections, implant retention is rarely successful. Implant removal leaves the patient disabled for weeks or even months [13]. Moreover, once a mature biofilm has developed, the infection cannot be cured without removing the implant [22,23]. Thus, the reason for the historically high failure rates of I&D with component retention alone could be explained by the persistence of biofilms in PJI.

Biofilms related infections are recalcitrant to antibiotic strategies. It has already been published in the literature that the antimicrobial concentrations needed to eradicate biofilms are higher than the concentrations required to eradicate the same bacterial clones in a planktonic state. To be able to decide on an appropriate empirical antimicrobial therapy, the common microbiological causes of periprosthetic joint infections should be known. In orthopedic surgeries, the most common causative organisms are Staphylococcus aureus and coagulase-negative staphylococci (CNS) [6]. In the recent report, Staphylococcus aureus was the most common isolated pathogen, followed by CNS in the causative microorganisms isolated in the last years from hip and knee periprosthetic joint infections [24]. The incidence of orthopedic methicillin-resistant Staphylococcus aureus (MRSA) infections has increased, and since it has also been proven effective against MRSA strains, vancomycin is recommended as the first-line antibiotic therapy choice for the treatment of orthopedic MRSA infections [25]. In this study, vancomycin hydrochloride (VCM) was used as the antibiotic for impregnation in the CHA in ten patients (11 hips). Antibiotic-impregnated CHA might be increased higher topical antimicrobial concentrations to eradicate biofilms.

Indications for determining eligibility for I&D have been suggested in recent years as the arthroplasty community has developed a greater understanding of the risk factors that predispose patients to treatment failure. Success can be achieved in over 70% of cases when patients with favorable factors are selected, such as those with a short duration of symptoms (less than 3–4 weeks), a stable implant, and healthy soft tissues surrounding the prosthesis [9,18,26]. For this reason, the Infectious Diseases Society of America published guidelines in 2013 recommending removal of the implant when PJI develops more than 30 days after the index arthroplasty [9].

In this retrospective study, we investigated the efficacy of antibiotic-impregnated CHA used in conjunction with irrigation and debridement for the treatment of PJI with attempted prosthetic retention for determination of long-term efficacy and safety. PJI could be treated successfully using antibiotic-impregnated CHA, even for more advanced cases. All patients with infective symptoms for less than 3 weeks were successfully treated with I&D using antibiotic-impregnated CHA. Moreover, seven of nine patients who had symptoms of infection for more than 3 weeks were successfully treated by I&D with antibiotic-impregnated CHA. Both of the two patients for whom I&D treatment failed had diabetes mellitus and symptoms of infection over 3 weeks. Supporting our results, Katakam A et al. have suggested that morbidly obese patients face an increased risk of DAIR failure [27].

The success of I&D in early PJI largely depends on the presence of a mature biofilm. Infected artificial joints are often unresponsive to antibiotic treatment owing to poor vascular supply and biofilm formation. In most cases, reconstruction was performed with cemented implants, and antibiotic-impregnated cement was used. At the time of revision surgery with the cemented prosthesis, the success rate with antibiotic-impregnated bone cement has been reported to be higher than that without antibiotic-impregnated bone cement [28]. However, during PJI surgery with a cementless prosthesis, a high concentration of antibiotics cannot be obtained around the prosthesis.

Previous studies, including our study, reported successful two-stage reconstruction surgery using CHA in patients with intractable PJI, with good clinical outcomes without any cases of reinfection during follow-up [29,30]. Importantly, our results showed that this novel antibiotic delivery system could be a useful tool for PJI surgery with cementless prosthesis. In a previous study by Sudo et al., antibiotic-impregnated CHA has been suggested as an approach for improving outcomes by providing a local antibiotic depot [29]. Shinto et al. first utilized CHA for the local delivery of antibiotics and demonstrated that antibiotic-impregnated CHA releases gentamicin sulfate, cefoperazone sodium, and flomoxef sodium for a longer period than antibiotic-loaded polymethylmethacrylate cement (ALAC) in vivo and in vitro [7]. Moreover, in vitro, gentamicin-impregnated CHA was reported to produce 2.5 times higher concentrations, for 1.2 times longer, than an ALAC drug delivery system [31]. Gentamicin-impregnated CHA ceramic was also shown to have the ability to deliver 5 times the minimum inhibitory concentrations for Staphylococcus species for at least 12 weeks. A recent in vitro study demonstrated that antibiotic-impregnated CHA releases active VCM for longer periods and in higher amounts than ALAC [30].

There were several limitations to the current study. First, this study was retrospective, and randomized trials are needed to conclusively determine whether the use of CHA could improve infection-free survival when used as part of an attempt at implant retention in the setting of PJI. Furthermore, because the study was retrospective in nature, it is possible that there may have been some selection bias owing to selection of CHA in more challenging cases. Second, in early PJI, there may be cases that can be treated with only I&D. However, recurrent infection may be a physical and mental burden on patients. Therefore, antibiotic-impregnated CHA can be used to lower the possibility of recurrence. Finally, our study cohort lacked a direct control group to compare irrigation and debridement with component retention with or without the addition of antibiotic-impregnated CHA.

## 4. Materials and Methods

### 4.1. Patients

This research was a retrospective study and was approved by the Institutional Review Board of our institution. Thirteen patients (14 hips) treated with I&D for PJI after THA at our institution between 1997 and 2017 were retrospectively enrolled in this study and followed for more than 2 years after treatment. The study group consisted of four men (five hips) and nine women, with an average age of 66.3 years (range, 56–90 years). The initial diagnoses were osteoarthritis in six patients, rheumatoid arthritis in four patients (five hips), idiopathic osteonecrosis in two patients, and neck fracture in one patient (Table 3).

The diagnosis of infection was based on clinical criteria, including the presence of discharging sinus, frank purulent fluid, or pus found on pre-operative hip aspiration or positive findings on laboratory and histopathological tests. The criteria to define PJI were the presence of a sinus tract communicating with the prosthesis and/or at least two identical positive cultures.

### 4.2. Evaluation of Outcomes

The primary outcome measure was the presence or absence of PJI at the most recent clinical follow-up and the final follow-up date. At the most recent follow-up visit, the patients were clinically assessed, and blood tests were performed, including analysis of erythrocyte sedimentation rate and C-reactive protein.

Functional outcomes were assessed based on the Japanese Orthopedic Association (JOA) hip score, with a maximum score of 100 points (representing no disability).

### 4.3. Success Criterion

The criterion for success was apparent resolution of the initial infection after a minimum follow-up of 2 years, defined as the absence of clinical, biological, and radiological implant infection signs or death directly related to the infection or treatment [32]. Treatment success was defined as the absence of infection after 2 years, with retention of the prosthesis.

## 5. Conclusions

In conclusion, our findings suggested that I&D treatment with antibiotic-impregnated CHA produced high success rates for the treatment of PJI after THA, even in cases of advanced disease, with adequate functional outcomes after surgery.

## Figures and Tables

**Figure 1 antibiotics-12-00938-f001:**
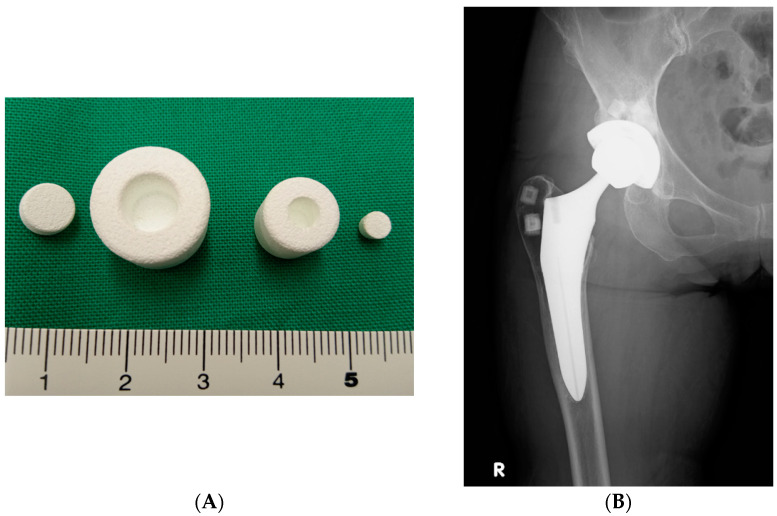
(**A**): Photograph of a calcium hydroxyapatite (CHA) ceramic block. (**B**): Radiographs of the right hip of a 64-year-old woman. Anteroposterior radiograph after revision total hip arthroplasty with CHA blocks placed into the major trochanter and acetabulum is shown. Radiographs of other cases are shown in Appendix A.

**Table 1 antibiotics-12-00938-t001:** Information for patients who underwent I&D with antibiotic-impregnated CHA treatment.

Case	Sex	Age	Diagnoses	Previous Surgery	Onset Symptom	Organism	Age of THA (Months)	The Time from Surgery to Onset Symptom	Antibiotics in the CHA Blocks (Numbers)	Antibiotics at Interim Period (Weeks)
1	M	65	ION	Hemi	Hip pain, fever	Escherichia coli	112	8 months	CTM, AMK (5)	CTM, AMK (6)
2	M	56	ION	THA	Hip pain, fever	Streptococcus agalactiae	8	3 months	FMOX (5)	FMOX, FOM, CTM, MINO, IPM/CS, PCG (5)
3	F	62	OA	THA	Chill	MRSA	31	3 months	VCM (2)	ABK, GM (9)
4	F	69	FX	Hemi	Hip pain	Staphylococcus epidermidis	1	1 month	VCM, CTM (5)	FOMX, ABK, AMK, VCM, TEIC (8)
5	F	52	OA	THA	Hip pain	Staphylococcus aureus	12	2 months	IPM/CS, CTM(4)	ABPC, CLDM, PIPC, IPM/CS (6)
6	F	71	OA	THA	Fever, fistula	CNS	1	2 weeks	VCM, AMK (3)	SBTPC, CTM, IPM/CS (8)
7	M	57	RA	THA	Hip pain	MSSA	78	3 days	VCM (2)	CEZ, MEPM, TEIC (7)
8	M	71	RA	THA(rt)	Hip pain	MSSA	43	2 weeks	VCM, FOM (3)	CEZ, PIPC, TEIC, LVFX, CLDM, RFP (6)
8	M	71	RA	THA(lt)	Hip pain	MSSA	36	2 weeks	VCM, FOM (3)	CEZ, PIPC, TEIC, LVFX, CLDM, RFP (6)
9	F	63	OA	THA	Hip pain, fever, fistula	MSSA	36	7 months	VCM, FOM (3)	CLDM, TEIC, VCM, RFP (7)
10	F	74	RA	THA	Swelling	E.coli	84	6 months	VCM, FOM (3)	CEZ, MEPM, IPM/CS, RFP (6)
11	F	68	OA	THA	Hip pain, fistula	MSSA	20	5 months	VCM, FOM (3)	CEZ, ABPC, ABPC/SBT, MINO, RFP (3)
12	F	64	RA	THA	Swelling	-	48	3.5 weeks	VCM (3)	CEZ, LVFX(6)
13	F	90	OA	THA	Hip pain, fistula	Enterococcus faecalis	57	2 weeks	VCM (2)	CEZ, LVFX(4)

I&D: irrigation and debridement, CHA: calcium hydroxyapatite, ION: idiopathic osteonecrosis of the femoral head, OA: osteoarthritis, FX: neck of femur fracture, RA: rheumatoid arthritis, Hemi: hemiarthroplasty, THA: total hip arthroplasty, CNS: coagulase-negative staphylococci, MSSA: methicillin-susceptible Staphylococcus aureus, MRSA: methicillin-resistant Staphylococcus aureus, CTM: cefotiam, AMK: amikacin, FMOX: flomoxef, VCM: vancomycin, MINO: minocycline, IPM/CS: imipenem/cilastatin, PCG: benzylpenicillin, ABK: arbekacin, GM: gentamicin, TEIC: teicoplanin, ABPC: ampicillin, CLDM: clindamycin, PIPC: piperacillin, SBTPC: sultamicillin, CEZ: cefazolin, MEPM: meropenem, RFP: rifampicin, ABPC/SBT: ampicillin/sulbactam, LVFX: levofloxacin.

**Table 2 antibiotics-12-00938-t002:** Clinical results of antibiotic-impregnated CHA for the treatment of periprosthetic joint infection.

Case	Reimplantation	Success/Failure	Treatment of Reinfection	Follow-up	Follow-up Periods after Treatment of PJI (Years)	Final JOA Score
1	I&D + exchange	Success		Died of other causes	5.7	
2	I&D	Success		Died of other causes	4.7	
3	I&D	Success		Regularly visits	18.6	49
4	I&D + exchange	Failure (Reinfection)	two-stage revision	Regularly visits	16.4	59
5	I&D + exchange	Failure (Reinfection)	two-stage revision	Died of other causes	12.5	46
6	I&D + exchange	Success		Regularly visits	11.6	42
7	I&D + exchange	Success		Regularly visits	5.3	93
8	I&D + exchange	Success		Regularly visits	9.0	77
8	I&D + exchange	Success		Regularly visits	9.0	78
9	I&D + exchange with cup	Success		Regularly visits	4.6	97
10	I&D + exchange	Success		Died of other causes	4.0	93
11	I&D + exchange with cup/stem	Success		Regularly visits	6.0	70
12	I&D + exchange	Success		Regularly visits	3.0	77
13	I&D + exchange	Success		Regularly visits	2.9	51

I&D: irrigation and debridement, CHA: calcium hydroxyapatite, PJI: periprosthetic joint infection.

**Table 3 antibiotics-12-00938-t003:** Demographics of patients who underwent I&D with antibiotic-impregnated CHA treatment.

Male 5 Hips (4 Patients) Female 9 Hips	
Osteoarthritis (OA)	6 hips
Rheumatoid arthritis (RA)	5 hips (4 patients)
Idiopathic osteonecrosis of the femoral head (ION)	2 hips
Neck of femur fracture (FX)	1 hip
Follow-up periods after treatment of PJI	Average 8.1 years (2.9~18.6)

I&D: irrigation and debridement, CHA: calcium hydroxyapatite, PJI: periprosthetic joint infection.

## Data Availability

Not applicable.

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
