# Peer review of "Outcome of Irrigation and Debridement with Topical Antibiotic Delivery Using Antibiotic-Impregnated Calcium Hydroxyapatite for the Management of Periprosthetic Hip Joint Infection"

_antibiotics, 2023, doi:10.3390/antibiotics12050938_

Round 1

Reviewer 1 Report

In my humble opinion, this well-written manuscript might be published after a minor revision. Namely:

1. lines 47-48: “Antibiotic-impregnated calcium hydroxyapatite (CHA) ceramic has recently been 47 developed as a new drug delivery system (Figure 1A) [6, 7] and has been used for …” – the term “recently” does not correlate with Refs. 6-7, because both of them were published in 1992.

2. Table 1: Some abbreviations appear to be unclear. The examples comprise ION, Hemi, OA, FX and some other ones. Please, re-check Table 1 and entire manuscript carefully and expand the meanings for all abbreviations.

Author Response

Replies to Reviewer #1

Authors are grateful to Reviewer #1 for encouraging comments. We have revised the indicated parts of the manuscript according to the comments. Corrections in the newly revised manuscript are underlined.

Please note that your review comments are shown in italic below and our replies in non-italic.

【General comments】

On the comment of [lines 47-48: “Antibiotic-impregnated calcium hydroxyapatite (CHA) ceramic has recently been developed as a new drug delivery system (Figure 1A) [6, 7] and has been used for …” – the term “recently” does not correlate with Refs. 6-7, because both of them were published in 1992.]

Reply: Thank you for your valuable suggestion. We deleted the term “recently”.

On the comment of [Table 1: Some abbreviations appear to be unclear. The examples comprise ION, Hemi, OA, FX and some other ones. Please, re-check Table 1 and entire manuscript carefully and expand the meanings for all abbreviations.]

Reply: Thank you for suitable suggestion. We revised abbreviations and added the following in Table 1. ION: Idiopathic osteonecrosis of the femoral head, OA: Osteoarthritis, FX: Neck of femur fracture, RA: Rheumatoid arthritis, Hemi: Hemiarthroplasty, THA: Total hip arthroplasty, CNS: coagulase-negative staphylococci, MSSA: meticillin-susceptible Staphylococcus aureus, MRSA: methicillin-resistant Staphylococcus aureus methicillin-resistant staphylococcus aureus

Reviewer 2 Report

• Scale should be added to Figure 1a and b.

• In the introduction part, the difference, advantages, and superiorities of the study from the literature should be emphasized.

• If there are characterizations performed on the “Antibiotic-impregnated calcium hydroxyapatite (CHA) ceramic” sample used in the study, they should be included in the manuscript.

• Line 112 should be checked.

• Table widths should be arranged to be the same.

• Sentences, tables, and contents similar to the studies in the literature should not be used. The work must be completely original. The manuscript should be checked thoroughly: https://doi.org/10.21203/rs.3.rs-842120/v1

• The current study should be compared by citing recent literature (especially 2023 and 2022).

Author Response

Replies to Reviewer #2

The authors are grateful to Reviewer #2 for insightful and critical review comments that significantly help improve the manuscript and increase clinical relevance. Corrections in the newly revised manuscript are underlined.

Please note that your review comments are shown in italic below and our replies in non-italic.

[General comment]

On the comment of [Scale should be added to Figure 1a and b.]

Reply: Thank you for your valuable suggestion. We added the scale in Figure 1A.

On the comment of [In the introduction part, the difference, advantages, and superiorities of the study from the literature should be emphasized.]

Reply: Thank you for suitable suggestion. As the difference, we added that “Chronic infections follow a persistent and undulating course with frequent exacerbations, are generally not fully responsive to systemic antibiotics, and often recur when treatment regimens are discontinued. Therefore, as the only strategy, topical and systemic antibiotics fail to successfully manage bacteria with a biofilm phenotype and must be combined with other approaches” on line 48-52. The advantages and superiorities were noted that long period and high rates of antibiotics topically in introduction on line 55-57. And we added that revision arthroplasty as I&D and two-stage using cementless prostheses can be used on novel strategies to control PJI” on line 57-58.

On the comment of [If there are characterizations performed on the “Antibiotic-impregnated calcium hydroxyapatite (CHA) ceramic” sample used in the study, they should be included in the manuscript.]

Reply: Thank you for suitable suggestion. However, we do not have data that we have characterized antibiotic-impregnated calcium hydroxyapatite.

On the comment of [Line 112 should be checked.]

Reply: Thank you for suitable suggestion. We revised that “The operative antibiotherapy was systematically initiated following pre-operative hip aspiration.”

On the comment of [Table widths should be arranged to be the same.]

Reply: Thank you for suitable suggestion. We arranged to be the same widths on each Table 1 and 2.

On the comment of [Sentences, tables, and contents similar to the studies in the literature should not be used. The work must be completely original. The manuscript should be checked thoroughly: https://doi.org/10.21203/rs.3.rs-842120/v1]

Reply: Thank you for suitable suggestion. We previously submitted to another journal and were rejected the manuscript. However, the manuscript been published as a preprint (not reviewed) on Research Square without our knowledge. And I checked with the editor, but it was a reply that there was no problem for submit.

On the comment of [The current study should be compared by citing recent literature (especially 2023 and 2022).]

Reply: Thank you for suitable suggestion. We cited and added the literature on 2022 (ref 6) and 2023 (ref 24).

Reviewer 3 Report

Review report

The manuscript presented a procedure to treat the infections caused by hip arthroplasty using antibiotic-containing apatite ceramics on a limited number of patients and then following up patients and evaluating their treatment trend. Some issues are required to explain before the manuscript may consider for publication. The details of the addressed comments as follows: 1.     The article has been submitted for publication in Antibiotic Journal. Therefore, it is expected that the reader will become more familiar with antibiotics for the application field proposed in this article. It is suggested that in the introduction of the article about the antibiotics used in this application, such as vancomycin or flomoxef etc., information on common microorganisms and effective antibiotics should be provided. Meantime, it is suggested to provide explanations in the introduction about two-stage revision arthroplasty.

2.     It is suggested to provide a radiographic image of the same hip that was infected and then treated by this method.

Author Response

Replies to Reviewer #3

The authors are grateful to Reviewer #3 for insightful and critical review comments that significantly help improve the manuscript and increase clinical relevance. Corrections in the newly revised manuscript are underlined.

Please note that your review comments are shown in italic below and our replies in non-italic.

 [General comment]

On the comment of [The article has been submitted for publication in Antibiotic Journal. Therefore, it is expected that the reader will become more familiar with antibiotics for the application field proposed in this article. It is suggested that in the introduction of the article about the antibiotics used in this application, such as vancomycin or flomoxef etc., information on common microorganisms and effective antibiotics should be provided. Meantime, it is suggested to provide explanations in the introduction about two-stage revision arthroplasty.]

Reply: Thank you for your valuable suggestion. We cited and added that “Biofilms related infections are recalcitrant to antibiotic strategies. It has been already published in the literature that the antimicrobial concentrations needed to eradicate biofilms are higher than the concentrations required to eradicate the same bacterial clones in a planktonica state. To be able to decide on an appropriate empirical antimicrobial therapy, the common microbiological causes of periprosthetic joint infections should be known. In orthopedic surgeries, the most common causative organisms are Staphylococcus aureus and coagulase-negative staphylococci (CNS). In the recent report, Staphylococcus aureus was the most common isolated pathogen, followed by CNS in the causative microorganisms isolated in the last years from hip and knee periprosthetic joint infections. The incidence of orthopedic methicillin-resistant Staphylococcus aureus(MRSA) infections has increased, and since it has also been proven effective against MRSA strains, vancomycin is recommended as the first-line antibiotic therapy choice for treatment of orthopedic MRSA infections. In this study, vancomycin hydrochloride (VCM) was used as the antibiotic for impregnation in the CHA in ten patients (11 hips). Antibiotic-impregnated CHA might be in-crease higher topical antimicrobial concentrations to eradicate biofilms” in discussion on line 204-218.

On the comment of [It is suggested to provide a radiographic image of the same hip that was infected and then treated by this method.]

Reply: Thank you for suitable suggestion. We added the radiographic image of other 5 cases in Supplementary Materials.

Reviewer 4 Report

I believe that this article is not suitable for publication in Antibiotics journal. The experimental design is not clear and is not detailed at all. Too much information is missing, including, for example, how the identification of the bacterial infections was assessed, and how the CHA was impregnated (concentration of the antibiotic in the solution, actual adsorption of the antibiotic on the CHA, previous in vitro tests?). If presented as Research Article, the text needs to be completely modified and improved with additional experimental data.

Author Response

Replies to Reviewer #4

The authors are grateful to Reviewer #4 for insightful and critical review comments that significantly help improve the manuscript and increase clinical relevance. Corrections in the newly revised manuscript are underlined.

Please note that your review comments are shown in italic below and our replies in non-italic.

[General comment]

On the comment of [I believe that this article is not suitable for publication in Antibiotics journal. ]

Reply: We have revised to respond your comments as well as other reviewers. Please review again.

On the comment of [The experimental design is not clear and is not detailed at all.]

Reply: Thank you for your valuable suggestion. We showed that “I&D for PJI after THA at our institution between 1997 and 2017 were retrospectively enrolled in this study and followed for more than 2 years after treatment”. We changed at first paragraph that “ This research was retrospective study and was approved by------” on line 274.

On the comment of [how the identification of the bacterial infections.]

Reply: Thank you for suitable suggestion. We showed on line 281-285 that “The diagnosis of infection was based on the clinical criteria, including the presence of discharging sinus, frank purulent fluid, or pus found on pre-operative hip aspiration or positive findings on laboratory and histopathological tests. The criteria to define PJI were the presence of a sinus tract communicating with the prosthesis and/or at least two identical positive cultures”.

On the comment of [how the CHA was impregnated.]

Reply: Thank you for suitable suggestion. We showed on line 116-119 that “We created bone holes on the major trochanter and the acetabulum using an air drill. Antibiotic-impregnated CHAs were implanted as often as possible when there were poor bone stocks because of severe bone loss. We implanted 2–5 antibiotic impregnated CHAs in bone holes.” And We added radiographic image of other 5 cases in which the implantation position of antibiotic-impregnated CHAs is easy to understand.

On the comment of [concentration of the antibiotic in the solution, actual adsorption of the antibiotic on the CHA, previous in vitro tests?]

Reply: Thank you for your valuable suggestion. However, we do not have data that we have characterized antibiotic-impregnated calcium hydroxyapatite.

Round 2

Reviewer 3 Report

All comments have addressed in new version of manuscript. It is suggested to publish.